# Quantifying Generalisation in Imitation Learning

**Nathan Gavenski**
Department of Informatics
King's College London
`nathan.schneider_gavenski@kcl.ac.uk`

**Odinaldo Rodrigues**
Department of Informatics
King's College London
`odinaldo.rodrigues@kcl.ac.uk`

## Abstract

Imitation learning benchmarks often lack sufficient variation between training and evaluation, limiting meaningful generalisation assessment. We introduce Labyrinth, a benchmarking environment designed to test generalisation with precise control over structure, start and goal positions, and task complexity. It enables verifiably distinct training, evaluation, and test settings. Labyrinth provides a discrete, fully observable state space and known optimal actions, supporting interpretability and fine-grained evaluation. Its flexible setup allows targeted testing of generalisation factors and includes variants like partial observability, key-and-door tasks, and ice-floor hazards. By enabling controlled, reproducible experiments, Labyrinth advances the evaluation of generalisation in imitation learning and provides a valuable tool for developing more robust agents.

## 1 Introduction

Imitation learning lies at the intersection of reinforcement and supervised learning. It can be seen as a relaxation of the reinforcement learning problem, where the agent learns a new skill through its own experiences, into a supervised learning setting, where the agent learns by observing others perform the same task. Like supervised learning, imitation learning relies on observed data for training. However, its agentic nature makes evaluation more akin to reinforcement learning, as the agent's performance is assessed through interactions with an environment rather than static comparisons against a dataset. As a result, many common evaluation benchmarks for imitation learning originate from the field of reinforcement learning. The most common benchmarks [8] for imitation learning are: (i) CartPole [1]; and (ii) MountainCar [14], which are classic control tasks; (iii) Ant [18]; and (iv) Humanoid [19], which are continuous control tasks; and (v) Atari Games, which set a benchmark for various games.

Classic control tasks, although reasonable for testing the initial capabilities of an imitation learning agent, are too simplistic to capture the complexities of real-world decision-making. They have low-dimensional state spaces and a limited range of discrete actions. As a result, they provide only a narrow evaluation of an imitation learning agent's capabilities. On the other hand, Continuous control tasks provide a more challenging evaluation setting. These environments feature high-dimensional state and action spaces, requiring agents to learn complex motor control strategies. However, they still share key limitations with classic control tasks, such as a lack of precise state abstractions and the expected behaviour for the agent at any given state. The first limitation refers to the vector state lacking information about the environment setting, such as the length of limbs for robots and the goal position, since the assumption is that the learned agent will be evaluated under the exact same constraints as it was during training. A common solution to incorporate this information is to use image-based states. However, when using images as states, the state may not accurately represent the difference between states due to the loss of precision from continuous numbers to pixel-based representation, and may exhibit partial observability since some parts of the agent may not be visible for the entire time. For the second limitation, in these environments, finding the optimal

39th Conference on Neural Information Processing Systems (NeurIPS 2025) Track on Datasets and Benchmarks.

expected behaviour for an agent in any given state is virtually impossible, which hinders the formal assessment of generalisation. Finally, Atari Games introduces diverse tasks with visual inputs and long-term strategic planning. While they provide a more varied and challenging benchmark, they remain constrained because the training data and test environments do not differ, meaning agents are evaluated under the same conditions in which they were trained. This prevents a clear separation between training and testing data, which is crucial for assessing generalisation.

To address these limitations, we introduce Labyrinth[1], a novel environment designed to: (i) explicitly separate training and test data by altering structure, goals, or starting positions, demanding generalisation; (ii) provide a discrete and fully observable state space, where all possible states, transitions, and optimal actions are explicitly defined, enabling precise analysis of an agent's decision-making; (iii) allow for the systematic analysis of an agent's ability to learn and adapt to structural changes, offering insights into its robustness and generalisation capabilities; and (iv) the environment can be easily customised to increase the difficulty further, e.g., by increasing the size of the labyrinth or maintaining the same solution set but changing the structure to analyse the inner parameters of the agent. Labyrinth offers a more robust and comprehensive benchmark for imitation learning, more effectively capturing the challenges of real-world learning scenarios that require drastic adaptation from the agent than existing environments.

## 2    Labyrinth Environment

In this work, we propose the Labyrinth environment to help assess the generalisation capabilities of imitation learning agents. Navigating through a labyrinth from designated starting and goal positions by observing the labyrinth's entire structure is a trivial task. Humans can find a route by analysing all paths connecting the start and goal positions, and then applying a given criterion to select one (e.g., the shortest). Classical problem-solving approaches, such as breadth-first search, can generalise to any labyrinth structure (considering the problem's solution and not its optimality). Therefore, navigating through a labyrinth should be considered an easy and well-suited task for measuring how well an imitation learning algorithm learns, and how general the agent's resulting capability is, i.e., by using structural configurations (e.g., wall locations, obstacles, etc) not present in the training data or moving from a different initial starting point.

Unlike other environments, a labyrinth offers some inherent characteristics: (i) agents cannot perform state-matching by forcing a path to be similar to its training data; (ii) changing the configuration of the labyrinth (walls, start and goal) does not affect the task and is easy to define; and (iii) changes between states are easier to identify since states can only differ by the agent's position. These characteristics allow us to perform a more systematic evaluation of different methods. For example, one can train a set of agents in one labyrinth structure and only the starting position or a subset of its walls. Alongside the traditional task of navigating a labyrinth to reach a goal, the Labyrinth environment offers the possibility of making solutions more complex by adding two additional components: (i) *key and door*, where the agent must retrieve a key to open a door before reaching the goal; and (ii) *ice floors*, where the agent must avoid stepping onto "frozen" (unsafe) tiles. We discuss the rationale for these two tasks and their importance to imitation learning in Sec. 2.3.

### 2.1    Structure and Actions

Labyrinth can create a new structure by specifying the desired number of rows and columns, including height and width. The coordinates of the starting ($s_0$) and goal tiles ($g$) can be either (i) user-defined – the user specifies where the start and goal tiles are located; (ii) biased – the starting tile is at the lower-left corner of the labyrinth and the goal is at its upper right, or (iii) unbiased – the goal and starting tiles are set randomly within the labyrinth, according to a minimum specified distance of each other (cf. the Manhattan distance $d(s_0, g) = \mid x_{s_0} - x_g \mid + \mid y_{s_0} - y_g \mid$). We refer to these as *biased* and *unbiased* due to the nature of the action distribution for all possible solutions in these structures (cf. Sec. 3). Biased structures will maintain the action distribution similar even when switching their structures, while unbiased ones will keep the distributions uniform for all actions, which will require the agent to focus more on each state instead of predicting the most likely actions.

---

[1]Source code available at: `https://github.com/NathanGavenski/Labyrinth`

It is easier to depict the structure of a labyrinth as a grid, but it is formally defined as a graph, where nodes represent tiles and edges represent connections between them (Figure 1). The graph we utilise is constructed by removing some edges, which in the visualisation is equivalent to adding a wall sectioning connections between two tiles. This graph representation allows us to quickly detect duplicates, find all possible solutions between start and goal nodes, and easily create configurations with different degrees of similarity to an existing labyrinth. Furthermore, configurations can be stored and subsequently reused or altered, allowing for the creation of datasets with specific characteristics and ensuring complete separation between training, validating, and testing sets.

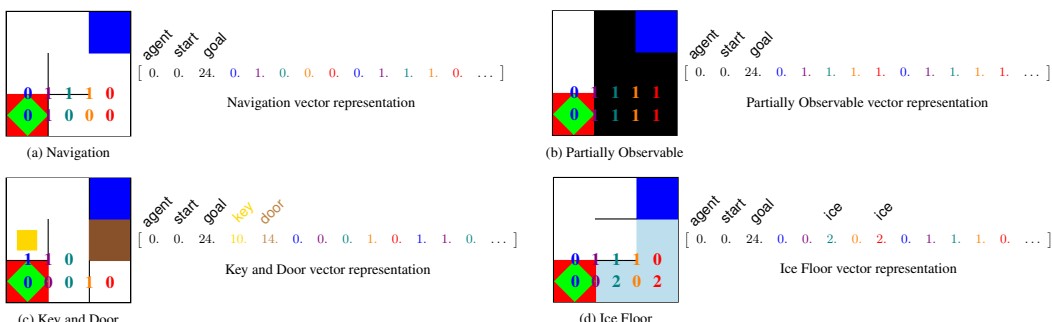

Figure 1: Different state representations for each task in the Labyrinth.

Even though the evaluation and test sets are distinct, we note two different features of this environment: (i) smaller labyrinths have higher chances of presenting similar trajectories to their goal (different structures while sharing a common path from $s_0$ to $g$; and (ii) the biased setting creates more straightforward solutions since the distribution of actions consists mainly of '*up*'and '*right*'actions.

The actions '*up*', '*down*', '*right*', and '*left*' move the agent one tile at a time towards the corresponding direction. It is important to note that the environment does not prevent an agent from taking an action towards a wall. However, in such cases, the agent's position will remain the same, although a unit of time will have elapsed. Finally, no actions can be executed once the goal tile is reached.

## 2.2   State and Reward

Each state consists of a labyrinth image with the agent drawn in its current position or a vector with the agent, start and goal global positions, and the labyrinth's structure. The start and goal tiles have different colours, red and blue, respectively, and the agent is a green diamond. Fig. 1a illustrates the default state of the labyrinth. Labyrinth can also return a partially observed state. This state consists of the tiles and walls in the immediate vicinity of the agent's current position.[2] It is important to note that we do not obfuscate start and goal positions since it would be impossible for the agent to know where these tiles are in the unbiased setting. In theory, partial observability would make it more complex for the agent to solve the environment. However, we hypothesise that since imitation learning tries to match the current state with a sample from the teacher, it could make it easier for the agent to reach the goal (even more so when considering the biased setting). Fig. 1b shows the partial state from the agent's perspective.

Even though imitation learning approaches ignore reward signals from environments in the learning process, we implement a reward function to differentiate the solution for each algorithm in our environment. For that, we use Eq. 1.

$$r_i = \begin{cases} \frac{-0.1}{width \times height} & \text{not at goal} \\ 1 + |\tau_s| \times \frac{0.1}{width \times height} & \text{at goal} \end{cases} \quad (1)$$

In Eq. 1, $|\tau_s|$ is the length of the shortest trajectory. It allows for the same reward independently of the labyrinth's structure. In other words, an agent that reaches the goal using the shortest path will always yield an accumulated reward of 1. Consequently, an average reward of 1 means the agent

---

[2]The source code at `https://github.com/NathanGavenski/Labyrinth/blob/main/src/labyrinth/utils/render.py#L157` contains the precise definition of the visibility settings

reached the goal in all episodes, which provides a fairer evaluation that is independent of the solution length. It is important to note that this reward function still gives the agent a positive reward when using a sub-optimal path as long as it does not roam endlessly. Nevertheless, the use of the reward function is not essential for our experimentation. If the agent learns to navigate the Labyrinth (i.e, understands how to avoid walls properly and manages to reach the goal in configurations not seen in the training data), we will consider that it has generalised successfully.

## 2.3 Settings

For the labyrinth, we consider four different settings: *labyrinth navigation*, where the agent has a typical labyrinth and will need to reach $g$ from $s_0$; *partially observable labyrinth*, where the agent has to reach the goal but only observes the structure close to it; *key and door*, where success requires the achievement of sub-goals in a specific sequence. For example, collect a key from tile $g_k$, before opening a door at tile $g_d$, to then be able to reach the final goal at tile $g$; and *ice floor*, where the agent must avoid frozen tiles.

**Labyrinth navigation**: offers a default setting for standard navigation training and evaluation. In the user-defined setting, researchers specify the position of $s_0$ and $g$ tiles. Alternatively, they can let the environment choose these positions according to the biased or unbiased settings (cf. Sec. 2.1). We believe biased settings are more straightforward for imitation learning agents to learn since they will keep the action distributions similar. Therefore, the agent must only learn to navigate different transition functions from new labyrinths. On the other hand, a possible evaluation setting for agents is keeping the structure of walls the same and only changing its initial position (same transition function, possibly different action distribution). Thus, this task allows for training and evaluation: (i) with different structures but with $s_0$ and $g$ always at the same tiles; (ii) with different structures and with $s_0$ and $g$ in different tiles; and (iii) with the same structures but with $s_0$ and $g$ in different tiles.

**Partially observable labyrinths**: changes the labyrinth navigation task only to display information close to the agent's position, $s_0$ and $g$ positions. We believe that using a partially observable environment might help the agent to focus on the relevant information. By observing the whole structure, the agent might consider a state out-of-distribution when a training sample may be similar from a local perspective. Using partially observable states does not remove the possibility of evaluating and testing an agent in the same conditions as the navigation setting. However, we consider changing the structure in an unbiased setting a more complex problem when partial observability is in place since the change in transition function with the out-of-distribution actions leads to a more diverse set of possible solutions.

**Key and door**: setting allows researchers to measure how well their imitation learning agent can learn a sub-task (collecting a key to open a door before reaching a goal). When creating a labyrinth with the key and door setting, the environment will first define the structure and then find possible positions for the key and door. To define the environment's structure, we only allow labyrinths with paths that share at least a tile from $s_0$ to $g$. Doing so avoids instances where the agent could reach $g$ without completing the sub-task. To define the door's position $g_d$, we find all possible paths from $s_0$ to $g$ and select the last shared tile among them. To define the key's position $g_k$, we also use all possible paths from $s_0$ to $g$ but select a random reachable tile from all tiles not present in the set. We select the last shared to ensure the maximum number of tiles possible for the key and select a tile not present in the set of solutions to ensure the agent did not collect the key by chance and that it was an intended decision from the agent. The key and door setting also allows for the same set of evaluations from the navigation setting with one additional evaluation where we keep the same structure and $s_0$, $g$ and $g_d$ positions and change only $g_k$'s position.

**Ice floor**: offers a setting for researchers to experiment with safety and generalisation problems. In this setting, if the agent steps on the ice, the tile will break, and the episode will terminate (fail). For this setting, the environment creates its structure and ensures that at least two possible solutions exist to reach $g$. We set this premise to guarantee that if we set one possible path with ice floors, there will be at least another path that will be safe for the agent to reach $g$. With all possible paths, we select one of the possible paths from the set of solutions and set the tiles to be ice. It is important to note that we only set the tiles unique to that path to avoid accidentally making all paths unsafe. During the evaluation, researchers can maintain $s_o$ and $g$ positions and the same structure but swap ice tiles from unsafe to safe paths.

## 2.4 Ease of Use, Reproducibility and Customisation

We understand that an environment must be easy to use, allow for customisation, and be reproducible for the community to adopt it. Therefore, we developed Labyrinth with all of these in mind. Labyrinth runs on 'gymnasium' [], allowing researchers who already use the highly adopted Python library to use the environment with minimal adaptation. A typical utilisation of Labyrinth is illustrated below:

```python
import gymnasium as gym
import labyrinth
environment = gym.make(
    "Labyrinth-v0", shape=(5, 5), occlusion=False,
    key_and_door=False, icy_floor=False, render_mode="rgb_array"
)
obs, info = environment.reset(options={"agent": True})
solutions = environment.solve(mode="all")
obs, reward, done, truncated, info = environment.step(action)
```

Line 2 registers the environment on gymnasium, Lines 3–6 define the environment, Line 7 creates a new environment and yields the first state, Line 8 provides all possible solutions for that Labyrinth, and Line 9 performs a random action in the environment, which returns the next state, reward, whether the agent arrived at $g$, whether the agent has fallen through an ice floor, and the environment's info. To define an instance of the environment the user has the following parameters: $shape$, which requires a tuple that defines the width and height; $occlusion$ sets the partially observable setting; $key\_and\_door$ enables the key and door setting; $icy\_floor$ enables the ice floors setting; and $render\_mode$ defines what type of state should the environment return (vector or image). It is important to note that $occlusion$, $key\_and\_door$ and $icy\_floor$ are mutually exclusive. The solver for the environment uses Johnson's algorithm [11] to find all possible paths from $s_0$ to $g$. Beyond all possible solutions, the solver also allows for the shortest solution, which will return a single solution, one of all possible shortest paths (when the structure has more than one path with the same length).

To ensure reproducibility, we allow users to save and load past instances as follows:

```python
from labyrinth.file_utils import convert_from_file, create_file_from_environment
create_file_from_environment(environment, "example.labyrinth")
environment.load(*convert_from_file("example.labyrinth"))
```

Line 2 saves the current setting of the environment to the file `example.labyrinth`, and Line 3 loads the file structure and setting in the current Labyrinth object. Therefore, a user can create a set of structures and settings for training and another for evaluation, keeping consistency between different training and evaluation cycles. In fact, Labyrinth provides a feature for the easy creation of these sets:

```
python -m labyrinth.generate --width 5 --height 5 --train 100 --eval 100 --test 100
```

where $train$, $eval$ and $test$ define the size of each set (100 in this example) and the $width$ and $height$ of the structure. We reiterate that Labyrinth ensures that each structure is unique by hashing its structure and controlling that each new structure is not present during the creation of all sets, i.e., each structure is unique in its set and among all sets.

```
key_and_lock: False
icy_floor: False
occlusion: False
labyrinth:
-------------
|   |     E |
|   +   + - |
|           |
|   + - +   |
| S |       |
-------------
end
```

To allow easy customisation of the environment structure, we create a custom setting language that enables users to visualise the structure of each file easily, but also allows for editing existing structures quickly. An example of this can be seen here: where Lines 1–3 define the settings for the environment and Lines 4–12 defines the structure. For defining the tile types, users can use $S$ for the first state $s_0$, $E$ for the goal $g$, $K$ and $D$ for key and door positions, respectively, and $I$ for setting ice tile positions.

Finally, we provide a set of labyrinths and data for training imitation learning agents on IL-Datasets [5], which hosts its datasets on HuggingFace [4]. IL-Datasets provides a convenient and uniform way to evaluate imitation learning methods and ensures that implementations are compared under the same conditions: seeds, training data and evaluation. Labyrinth can be used without IL-Datasets, it is

used for its convenience and the benefits it provides to researchers. We create datasets for squared labyrinths with sizes of 3, 4, and 5. Each dataset consists of three splits (train, evaluation, and test), each split consisting of the shortest paths from $s_0$ to $g$ on the biased setting. Each dataset entry consists of the image observation, action, immediate reward, whether that entry is the first for an episode and the labyrinth information for recreating the same experiment. If users desire to use the unbiased setting, they can load the information from each entry and change $s_0$ and $g$ positions by using functions *change_start_and_goal* and *change_start*.

## 3    On the Generalisation Requirements for Benchmarks

For testing generalisation, we believe an environment needs some key requirements: (i) poses a challenging task; (ii) a significant change from training to evaluation; (iii) controls over these changes; and (iv) allows for debugging of the agent.

We argue that the task must be non-trivial for the first requirement and demand reasoning beyond memorisation. Labyrinth addresses this by requiring agents to plan long-horizon, reason over topological structures, and adapt to altered starting and goal states. Moreover, its variants, such as key-and-door and icy floor settings, add complexity through temporal dependencies and safety constraints, respectively. These extensions prevent shortcut solutions and promote learning robust decision-making strategies. Unlike classical benchmarks, where solutions can often be reduced to reactive policies, solving Labyrinth consistently demands trajectory-level reasoning and adaptation.

For the second requirement, we analyse the Labyrinth environment and the most common environments used in imitation learning benchmarks [8]: MountainCar, CartPole, Hopper, Walker-2D and HalfCheetah. Table 1 shows $100,000$ different initial states for the most common environments. For it, we initialise the environment with a seed not used for generating the training dataset, and use the closest average distance, based on the Manhattan distance, to it. We observe that most initial states are quite similar to the training data. This is not ideal since, by having states that are closer to the training data, imitation learning agents can adopt a behaviour-seeking mode, where the agent tries to use the expert's action instead of predicting the most adequate action for a given state. Ideally, the reward functions in environments would account for these less-than-optimal actions and show divergence in the behaviour. However, when doing this analysis, we encounter a significant downside of these environments. For CartPole and MountainCar, we could reach results comparable to those of the expert by recording a single sequence of expert actions and repeating it in a new initialisation. For example, for the MountainCar environment, a classical environment with a more challenging dynamic (agents have to build up momentum to reach the goal), we record an episode of accumulated reward of $-106.45$. By simply using the same sequence of actions over 100 different episodes, we reach an average accumulated reward of $-104.87 \pm 0.8562$. It is important to note that MountainCar consider the task solved when the agents achieve an average accumulated reward of $-110$. Yet, classical environments are considered simplistic in nature, as pointed out in Sec. 1. Therefore, we also analyse how these continuous tasks perform under different initialisations. In it, we discover that these environments are quite lenient over the actions taking place. For example, on the Hopper environment, by retrieving the closest state from the current environment one on a different seed and performing the exact expert action from the training data, we achieve a reward of $3530.2367 \pm 15.5748$, while the expert achieves $3536.3626 \pm 9.5699$, a marginally better result. These results are worrisome since most imitation learning works use these benchmarks to show that their model learned the underlying task and can generalise well to other initialisations.

Table 1: Manhattan distance for $1e5$ initialisations for the Gym and DeepMind control suites.

| Environment | Gym | | DeepMind | |
|---|---|---|---|---|
| | Summation | Average | Summation | Average |
| MountainCar | $0.0021 \pm 0.0016$ | $0.0010 \pm 0.0070$ | - | - |
| CartPole | $0.0380 \pm 0.0204$ | $\mathbf{0.0095 \pm 0.0051}$ | $\mathbf{0.0932 \pm 0.0189}$ | $0.0093 \pm 0.0047$ |
| Hopper | $2.3931 \pm 0.0091$ | $0.2175 \pm 0.0008$ | $3.5610 \pm 0.4974$ | $0.3237 \pm 0.0452$ |
| Walker-2D | $5.4045 \pm 0.0117$ | $0.3179 \pm 0.0060$ | $8.3509 \pm 0.6852$ | $0.4912 \pm 0.0403$ |
| HalfCheetah | $\mathbf{12.5915 \pm 0.3289}$ | $\mathbf{0.7406 \pm 0.0193}$ | $12.3376 \pm 0.2032$ | $0.7257 \pm 0.0119$ |

To understand how the labyrinth diverges from training, we analyse the action distribution over all possible settings (described in Sec. 2.3). Figure 2 shows the cell distribution and action distributions

for the solutions over the train, evaluation, and test splits for a $5 \times 5$ labyrinth[3]. We observe that for the first two settings (Fig. 2a and 2b) the action distribution remains close to the same during each split. However, the cell distribution changes, which means that to reach $g$, the agent will have to adapt its solution and better rank information to achieve the goal. In other words, if the agent only learns to find the closest state to the training data, and perform the same action, there will be labyrinth settings it will not solve. Moreover, by changing both $s_0$ and $g$ and maintaining the same structure (Fig. 2c), the action distribution drastically shifts to a more uniform one.

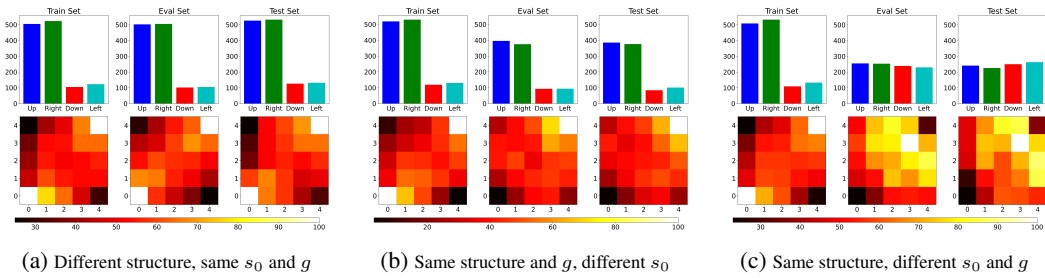

(a) Different structure, same $s_0$ and $g$     (b) Same structure and $g$, different $s_0$     (c) Same structure, different $s_0$ and $g$

Figure 2: Tile and action distribution over different settings for the Labyrinth environment.

The third requirement is necessary for all methods to be evaluated under fair and explainable conditions. Labyrinth enables precise control over how environments differ between training and evaluation, allowing researchers to isolate specific generalisation challenges. For instance, one can hold the labyrinth structure fixed while varying the agent's initial position (Fig. 2b), or conversely, alter the structure while maintaining consistent start and goal locations (Fig. 2a). This granularity helps identify whether failure modes are due to perceptual mismatch, action distribution shifts, or poor task abstraction. Full access to the graph structure makes it easier to inspect agent failures and identify brittle behaviour, which is often opaque in high-dimensional or continuous control settings.

The final requirement is that the environment must allow for effective debugging and inspection of the agent's behaviour. Labyrinth satisfies this by offering full access to both the structural and observational components of the environment, and by allowing researchers to place the agent in any arbitrary state. More importantly, due to its discrete and fully defined transition graph, Labyrinth allows us to compute the optimal action for every individual state under any configuration. This enables researchers to directly test whether the agent selects the correct action in a given state, quantify deviations from optimal behaviour, and compare across structurally similar settings. In contrast, widely used benchmarks such as MuJoCo-based (e.g., Hopper, HalfCheetah) and Atari-based environments make it virtually impossible to define the optimal action in most states due to high-dimensional, continuous dynamics and implicit goals. Similarly, in visual environments like Atari, researchers may not even have access to the full internal state, making it difficult to determine what constitutes a correct action. Labyrinth's explicit structure and ground-truth optimality afford a level of transparency and controllability that these environments lack, making it especially suitable for interpretability, policy debugging, and fine-grained evaluation of generalisation.

## 4 Benchmarking common imitation learning methods

We now benchmark some imitation learning methods to demonstrate the effectiveness of this environment in testing generalisation. Due to space constraints, this section only displays the results for the *labyrinth navigation* setting. We show all other settings in the supplementary material.

### 4.1 Implementations and Metrics

We use implementations from IL-Datasets [5] for all imitation learning methods. IL-Datasets provides us with implementations for Behavioural Cloning (BC) [15], DAgger [17], Generative Adversarial Imitation Learning (GAIL) [10], Behavioural Cloning from Observation (BCO) [20], Soft Q Imitation Learning (SQIL) [16], and Imitating Unknown Policies via Exploration (IUPE) [6]. We selected these methods because they offer a diverse range of imitation learning approaches. BC, GAIL, DAgger,

---

[3]The supplementary material contains all other labyrinth sizes.

and SQIL are all imitation learning from demonstration methods, while BCO and IUPE are imitation learning from observation methods. Moreover, DAgger requires access to the expert, which can benefit the training since any given labyrinth knows the optimal action. On the other hand, GAIL, BC and BCO are offline (do not interact with the environment during training), and the others are online. To prevent any method from accessing other labyrinth structures outside of those in the training data, we enforce that the online portion of their training is conducted only under the same conditions as those in which the dataset was created. In other words, we load the same labyrinth structures from the training dataset split during these interactions. Finally, they are also diverse in their learning approaches, employing adversarial and inverse reinforcement learning (GAIL, DAgger, and SQIL), dynamic methods [8] (BCO and IUPE), or behavioural cloning (BC).

In this work, we use two metrics: *average episodic reward* ($AER$) and *success ratio* ($SR$). $AER$ is the average reward the agent accumulates over $n$ episodes. In our experiments, we display the $AER$ for each of the 100 train, evaluation, and test labyrinths. An $AER$ of 1 means the agent achieves the goal using the shortest path. $SR$ is the ratio of the agent achieving the goal tile over $n$ episodes.

## 4.2 Results

Table 2 shows the benchmark results for each method in a $5 \times 5$ labyrinth with the same starting and goal tiles (biased setting) for the training, evaluation, and test splits. The dataset for this benchmark[4] (and for all others in the supplementary material) is hosted on HuggingFace [4], as explained in Section 2.4, and we provide all links to the datasets used in this work in the supplementary material. Besides the images, the dataset also contains all the information needed to recreate each labyrinth according to each entry. The experiments in Tab. 2 use the convolutional neural network based on the original Atari Deep Q-Network [13] as the encoder for each model, and we train all methods for $1,000$ epochs. As an addendum, we conducted additional experiments for other labyrinth sizes and settings, and a brief ablation of other neural network structures. These are described in the supplemental material.

Our experiments show that pure imitation learning methods (those not using inverse reinforcement learning techniques) perform better in Labyrinth. We believe that the reason for this is that pure imitation learning methods rely primarily on supervised learning losses, which encourages these models to learn better encodings for each image state. This results in the model generalising more, i.e., performing better in labyrinth structures not seen during training. When looking for the closest training examples in the encoding space given an evaluation or a test input, we discover that the agent's position itself for these models is less important than the actual wall structure surrounding the position. In these cases, the closest training images might have the agent in a different position, but the wall structure remains similar. The inverse reinforcement learning methods' optimisation is less direct, and the models do not learn the same patterns, resulting in less optimal behaviour. Unfortunately, all methods perform poorly in this setting, except IUPE, which is the only method to achieve a result higher than $10\%$ on the evaluation set. Yet, this result did not translate into the test split, which we see as evidence that IUPE did not learn the navigation task itself. It generalised well in the validation set, but not in the testing one. The other methods performed similarly badly in the test and validation sets.

Table 2: Benchmark results for training, validation and testing splits.

| Splits | Metric | BC | DAgger | GAIL | BCO | SQIL | IUPE |
|--------|--------|-----|--------|------|-----|------|------|
| Train | $AER$ | $-2.11 \pm 2.41$ | $-1.18 \pm 2.45$ | $-0.98 \pm 1.89$ | $-0.53 \pm 2.23$ | $-3.80 \pm 0.96$ | $\mathbf{0.27 \pm 2.39}$ |
| | $SR$ | 37% | 57% | 61% | 70% | 4% | **75%** |
| Valid. | $AER$ | $-3.70 \pm 1.18$ | $-3.75 \pm 1.08$ | $-3.57 \pm 1.58$ | $-3.90 \pm 0.69$ | $-3.95 \pm 0.49$ | $\mathbf{-2.80 \pm 2.12}$ |
| | $SR$ | 6% | 5% | 9% | 2% | 1% | **21%** |
| Test | $AER$ | $-3.90 \pm 0.70$ | $-3.80 \pm 0.97$ | $-3.85 \pm 0.85$ | $-3.85 \pm 0.85$ | $-4.00 \pm 0.00$ | $\mathbf{-3.85 \pm 1.00}$ |
| | $SR$ | 2% | 4% | 3% | 3% | 0% | **5%** |

Finally, to understand whether Labyrinth was too complex a challenge for imitation learning, we evaluated BC under an extended period of training ($10,000$ epochs) and using a more robust neural network architecture (ResNet-18 [9]). We chose BC as the baseline for this experiment because it is the most simplistic approach to pure imitation learning and the worst-performing of these methods.

---

[4]`https://huggingface.co/datasets/NathanGavenski/Labyrinth-v0_5x5`

When running BC with the same structure but for an extended period, it achieved $100\%$ during training, $41\%$ in the validation, and $34\%$ in the test splits, an improvement over Tab. 2 results. Yet, when running BC with a ResNet encoder, it achieves $100\%$ $SR$ in the training, $56\%$ in the validation and $53\%$ in the test splits, a significant improvement from the results in Tab. 2. We believe these numbers result from the model learning a less spurious encoding space. Therefore, with time, the model learns the correct characteristics to classify the correct action. However, it does not learn how to perform the underlying task of navigating the structure. This is backed up by the fact that improving the encoding architecture improves the method's performance, but does not guarantee the results from the other splits.

In summary, our experiments highlight that existing imitation learning methods struggle to generalise effectively in the Labyrinth environment, especially when faced with unseen structures. These results show Labyrinth's suitability for rigorous generalisation testing and underscore the need for more robust learning approaches.

## 5   Conclusion

In this work, we proposed Labyrinth, an easy-to-use, reproducible, and customisable environment for testing generalisation with imitation learning agents. Labyrinth provides researchers with: (i) a way to explicitly separate training, validation and test data via different labyrinth structures, start and goal positions; (ii) a discrete and fully observable state space where all possible states, transitions and optimal actions are explicitly defined, enabling precise analysis of an agent's decision-making process; (iii) a way to systematically analyse an agent's ability to learn and adapt to structural changes and action distribution shifts, offering insights into the agent's robustness and generalisation capabilities; and (iv) the ability to increase difficulty while preserving the nature of the task, and to analyse the inner parameters of the agents.

We analysed other commonly used imitation learning benchmarks and showed how the field could benefit from using Labyrinth as a platform for testing generalisation. Labyrinth is challenging enough to require agents to learn the underlying task to solve each unseen labyrinth structure. It offers customisable evaluation sets that are different enough from the training data (e.g., action distribution shift and other transition functions) to allow for controlled evaluation and debugging of each agent. Furthermore, Labyrinth provides the same features as all other standard benchmarks, such as accessibility via gymnasium, vector and image representations, and a reward function to compare different agents' results.

We performed a benchmark in the Labyrinth environment using common imitation learning methods, concluding that the field has yet to improve its generalisation capabilities. Although machine learning techniques can improve their results when solving unseen structures, they still do not generalise well, even if the action distribution remains the same. Moreover, the type of generalisation from the machine learning field would not theoretically apply to the required generalisation for the agents in this setting. To achieve a high success rate across each split, agents must build knowledge for the underlying task (navigation) instead of only correlating training samples to the agent's current state. We believe Labyrinth can help researchers benchmark their method's generalisation capabilities and improve the field perception over how to benchmark novel methods better.

Finally, Labyrinth comes with some limitations we envision tackling in the future. As it is developed now, Labyrinth only allows for discrete actions, which is ideal for finding the optimal action for each state. However, some imitation learning methods are only suited for continuous actions, such as OPOLO [21], MAHALO [12] and CILO [7]. Ideally, we would like to provide the option of performing continuous actions while keeping all the features Labyrinth provides (cf. 2), which other labyrinth-like environments do not have (such as Ant and Point Maze [2]). We would also like to develop a customisable tile feature that would allow researchers to specify particular behaviours in some tiles easily. As it stands now, this can be done, but requires researchers to change the source code in the environment.

## Acknowledgments and Disclosure of Funding

This work was supported by UK Research and Innovation [grant number EP/S023356/1], in the UKRI Centre for Doctoral Training in Safe and Trusted Artificial Intelligence (`www.safeandtrustedai.org`) and made possible via King's Computational Research, Engineering and Technology Environment (CREATE) [3].

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
