# Supplementary Material for Quantifying Generalisation in Imitation Learning

**Nathan Gavenski**
Department of Informatics
King's College London
nathan.schneider_gavenski@kcl.ac.uk

**Odinaldo Rodrigues**
Department of Informatics
King's College London
odinaldo.rodrigues@kcl.ac.uk

## 1   Datasets

Labyrinth provides three datasets, for three different sizes: (i) $3 \times 3$; (ii) $4 \times 4$; and (iii) $5 \times 5$. All datasets consist of at least one solution for 100 different labyrinths, with the exception of $3 \times 3$, which has 32 labyrinths per split. Each split is entirely unique, and no labyrinth appears more than once in the entire dataset, regardless of its split. Each entry consists of five pieces of information:

- *obs*: the path to the image representation for that state;
- *actions*: the integer action performed for that solution in that state;
- *rewards*: the float reward received for that action in that state;
- *episode_starts*: the boolean status that states if it is the first state in an episode; and
- *info*: the textual information required to load the same labyrinth structure if needed.

We provide images in each dataset since we believe that visual information is more useful to the imitation learning agent, and if a vector representation is needed, the *info* parameter allows researchers to load the same structure and the *actions* enables the recreation of the dataset in its vector format.

Each observation is an image of size $600 \times 600 \times 3$. Although each baseline trained in this work uses a $64 \times 64 \times 3$ input, we thought that providing a bigger image would benefit models requiring downsizing (e.g., the walls will not disappear during resizing). We note that the dataset has the last state in each episode (when the agent is at the goal $g$). Therefore, some entries might not be helpful to the agent. We recommend researchers to split entries intro episodes (since they are in sequence), and create tuples $(s, a)$ or $(s, a, s')$, where $s$ is the state, $a$ is the action, and $s'$ is the state resulting from the transition function given $s$ and $a$. IL-Datasets' [4] 'BaselineDataset' provides this implementation as an example. All datasets can be found in the HuggingFace collection at: `https://huggingface.co/collections/NathanGavenski/labyrinth-datasets-68245a55019fed6983502805`.

## 2   Configuration Language

As explained in Sec. 2.4 of the main work, Labyrinth uses its own configuration language to save and load existing structures. We create this language to support researchers when manually editing a labyrinth. Initially, we stored each labyrinth by writing the edges of the graph that models the structure (cf. Sec. 2.1 of the main work). However, modifying the file required a lot of effort since edges are stored by their global location (the inline position instead of their $x$ and $y$ coordinates). Therefore, we envisioned a more friendly visualisation that would help researchers modify the structure or the task without the need for a deep understanding of the labyrinth's inner structure.

This language supports saving and loading all possible configurations (as displayed in Fig. 1). Lines 1-3, define the setting for the labyrinth and are mutually exclusive. The Labyrinth also only loads tiles relevant to the setting. In other words, if all parameters are 'False' but there are ice tiles (such

as in Fig. 1d in Lines 4-12), it will not load any ice tiles. Yet, for convenience, if the labyrinth is instantiated in one setting, Fig. 1c for example, and afterwards, a file with a different configuration is loaded, such as Fig. 1d, the labyrinth will change to load the file in its entirety. In other words, it will overwrite the current setting ('key and door') and load the new setting ('ice floors') with the new tiles.

```
1 key_and_lock: False    1 key_and_lock: False    1 key_and_lock: True     1 key_and_lock: False
2 icy_floor: False        2 icy_floor: False        2 icy_floor: False        2 icy_floor: True
3 occlusion: False        3 occlusion: True         3 occlusion: False        3 occlusion: False
4 labyrinth:              4 labyrinth:              4 labyrinth:              4 labyrinth:
5 ------------            5 ------------            5 ------------            5 ------------
6 |   |      E |          6 |   |      E |          6 |     | D   E |          6 |   |      E |
7 |   +   + - |          7 |   +   + - |          7 |   +   + - |          7 |   +   + - |
8 |           |          8 |           |          8 |           |          8 |         I |
9 |   + - +   |          9 |   + - +   |          9 |   + - +   |          9 |   + - +   |
10 | S |       |         10 | S |       |         10 | S | K     |         10 | S   I   I |
11 ------------           11 ------------           11 ------------           11 ------------
12 end                    12 end                    12 end                    12 end
```

|     (a) Navigation     |  (b) Partial Observability  |  (c) Key and Door  |  (d) Ice Floor  |

Figure 1: Save files for all possible settings.

Lines 4-12 define the structure for each labyrinth. They define vertical walls by using pipes 'l', horizontal ones by hyphens '-', the start position $s_0$ as 'S', the goal $g$ as 'E', and the point between all four tiles by a plus sign '+'. Researchers can define any navigation labyrinth by using these symbols. It is important to note that the configuration language requires the outer walls of the labyrinth to know the size (height and width) of the labyrinth trying to be loaded. Additionally, if the user wants to define the key and door positions ($g_k$ and $g_d$, respectively) they can do so by using 'K' for key and 'D' for door. For loading ice in a specific tile, researchers can use 'I' for it. We note that if researchers define a configuration that has one of these settings turned on but does not define any location for the special tiles, the Labyrinth will define them using its internal heuristics (cf. Sec. 2.3 of the main work). Finally, the configuration supports multiple- and single-line comments by using '"""'. So, if researchers want to comment something in the file, they can do so by opening and closing their comment with three quotation marks.

# 3 Experimental Information

In this work we run all experiments using a machine with A100 NVidia GPU and 32GB of memory. However, the networks are relatively small, and these settings are not required to reproduce any of the results available here or in the main work. All methods are provided by IL-Datasets [4], which uses PyTorch and Adam's optimiser to update the weights. As for learning rates, we use the default value $5 \times 10^{-4}$ for all methods.

## 3.1 Network Topology

In the main work, we experiment with six different imitation learning methods (see Sec. 4 of the main work). For the experiments in Tab. 2 of the main work, we use the original encoder from Deep Q-Network [1] (Tab. 1a) and later with a ResNet-18 [6] (Tab. 1b). These structures are displayed in Tab. 1. We note that for the ResNet neural network, we follow PyTorch's implementation available at: `https://pytorch.org/hub/pytorch_vision_resnet/`, and the classifier layers remain the same for both encoders.

Table 1: Neural network topology for encoders.

| N | (a) CNN | | | (b) Resnet-18 | | |
|---|---|---|---|---|---|---|
| | Layer Name | Input × Output | Extra Info. | Layer Name | Input × Output | Extra Info. |
| 1 | Conv2d | $3 \times 32$ | Kernel: 8 Stride: 4 | Conv2d | $3 \times 64$ | Size: 7 Stride:2 Padding: 3 |
| 2 | LeakyReLU | - | Slope: 0.01 | BatchNorm2d | $64 \times 64$ | Mom.: 0.1 Affine: True |
| 3 | Conv2d | $32 \times 64$ | Kernel: 4 Stride: 2 | ReLU | - | - |
| 4 | BatchNorm2d | $64 \times 64$ | Mom.: 0.1 Affine: True | Max Pool | - | Size: 3 Stride: 1 Padding: 1 |
| 5 | LeakyReLU | - | Slope: 0.01 | ResNet Block ($4\times$ – Start) | | |
| 6 | Conv2d | $64 \times 64$ | Kernel: 3 Stride: 1 | Conv2d | $64 \times 64$ | Size: 3 Stride: 1 Padding: 1 |
| 7 | BatchNorm2d | $64 \times 64$ | Mom.: 0.1 Affine: True | BatchNoorm2d | $64 \times 64$ | Mom.: 0.1 Affine: True |
| 8 | LeakyReLU | - | Slope: 0.01 | ReLU | - | - |
| 9 | Fully Connected | $1024 \times 512$ | - | Conv2d | $64 \times 64$ | Size: 3 Stride: 1 Padding: 1 |
| 10 | LeakyReLU | - | Slope: 0.01 | BatchNorm2d | $64 \times 64$ | - |
| 11 | Dropout | - | Prob.: 30% | ResNet Block ($4\times$ – End) | | |
| 12 | Fully Connected | $512 \times 512$ | - | Fully Connected | $512 \times 512$ | - |
| 13 | LeakyReLU | - | Slope: 0.01 | LeakyReLU | - | Slope: 0.01 |
| 14 | Dropout | - | Prob.: 30% | Dropout | - | Prob.: 30% |
| 15 | Fully Connected | $512 \times 4$ | - | Fully Connected | $512 \times 512$ | - |
| 16 | | | | LeakyReLU | - | Slope: 0.01 |
| 17 | | | | Dropout | - | Prob.: 30% |
| 18 | | | | Fully Connected | $512 \times 4$ | - |

Finally, for GAIL's discriminator, we use a simple visual encoder (displayed in Tab 2).

Table 2: Discriminator Topology

| N | GAIL's Discriminator | | |
|---|---|---|---|
| | Layer Name | Input × Output | Extra Info. |
| 1 | Conv2d | $3 \times 32$ | Size: 8 and Stride: 4 |
| 2 | LeakyReLU | - | Slope: 0.01 |
| 3 | Conv2d | $32 \times 64$ | Size: 4 and Stride: 2 |
| 4 | LeakyReLU | - | Slope: 0.01 |
| 5 | Conv2d | $64 \times 64$ | Size: 3 and Stride: 1 |
| 6 | LeakyReLU | - | Slope: 0.01 |
| 7 | Fully Connected | $1028 \times 256$ | - |
| 8 | LeakyReLU | - | Slope: 0.01 |
| 9 | Fully Connected | $256 \times 256$ | - |
| 10 | LeakyReLU | - | Slope: 0.01 |
| 11 | Fully Connected | $256 \times 1$ | - |

 # 4 Extra Benchmark Results

This section provides a detailed breakdown of agent performance across increasingly complex Labyrinth sizes. By isolating each configuration, we assess how well imitation learning methods generalise beyond the training distribution as structural diversity and trajectory length increase.

## 4.1 Results for $3 \times 3$ Labyrinths

We expect labyrinths that are $3 \times 3$ of size to be an easier challenge to imitation learning methods. As stated in the main work in Sec. 2.1, labyrinths of smaller size have higher chances of presenting similar trajectories to their goal. Therefore, imitation learning methods, even with smaller encoders, are expected to achieve higher $SR$.

Table 3: Results for training, validation and testing splits for the navigation task in a $3 \times 3$ Labyrinth.

| Splits | Metric | BC | DAgger | GAIL | BCO | SQIL | IUPE |
|---|---|---|---|---|---|---|---|
| Train | $AER$ | $-1.36 \pm 2.48$ | $-0.74 \pm 2.35$ | $-1.03 \pm 2.45$ | $\mathbf{-0.59 \pm 2.30}$ | $-2.50 \pm 2.27$ | $-1.22 \pm 2.45$ |
| | $SR$ | $53.12\%$ | $65.62\%$ | $59.38\%$ | $\mathbf{68.75}\%$ | $31.25\%$ | $56.25\%$ |
| Valid. | $AER$ | $\mathbf{-0.59 \pm 2.30}$ | $-1.06 \pm 2.43$ | $-1.50 \pm 2.5$ | $-1.36 \pm 2.48$ | $-3.24 \pm 1.77$ | $-0.91 \pm 2.39$ |
| | $SR$ | $\mathbf{68.75}\%$ | $59.38\%$ | $50.00\%$ | $53.12\%$ | $15.62\%$ | $62.50\%$ |
| Test | $AER$ | $\mathbf{-0.74 \pm 2.36}$ | $-1.20 \pm 2.45$ | $-1.22 \pm 2.45$ | $-0.90 \pm 2.40$ | $-2.93 \pm 2.02$ | $-0.92 \pm 2.39$ |
| | $SR$ | $\mathbf{65.62}\%$ | $56.25\%$ | $56.25\%$ | $62.50\%$ | $21.87\%$ | $62.50\%$ |
| $s_0$ | $AER$ | $-1.20 \pm 1.65$ | $-1.36 \pm 2.48$ | $-0.87 \pm 2.42$ | $-1.04 \pm 2.44$ | $-2.31 \pm 2.27$ | $\mathbf{-0.45 \pm 2.23}$ |
| | $SR$ | $56.25\%$ | $53.12\%$ | $62.50\%$ | $59.38\%$ | $34.37\%$ | $\mathbf{71.87}\%$ |
| $s_0$ and $g$ | $AER$ | $-3.38 \pm 1.64$ | $-3.69 \pm 1.20$ | $-3.37 \pm 1.65$ | $-3.22 \pm 1.80$ | $\mathbf{-2.47 \pm 2.27}$ | $-3.69 \pm 1.20$ |
| | $SR$ | $12.50\%$ | $6.25\%$ | $12.50\%$ | $15.62\%$ | $\mathbf{31.25}\%$ | $6.25\%$ |

Table 3 shows that a smaller labyrinth size indeed facilitates imitation learning. The pure imitation learning supervised approaches achieve the highest $SR$. SQIL consistently underperforms across splits, with the exception of unseen conditions ($s_0$ and $g$), where it achieves $31.25\% \ SR$. IUPE demonstrates the highest success rate on $s_0$, suggesting better robustness in generalisation to novel initial states, but drastically underperforms on different initial states and goals. We hypothesise that the shift over the actions (to a more uniform one, as stated in Sec. 3) is too drastic for these methods to generalise and achieve the goal. These results highlight that while simple environments boost overall performance, generalisation remains a core challenge, with only a subset of methods, notably IUPE and BCO (self-supervised imitation learning methods), showing promise beyond the training distribution.

## 4.2 Results for $4 \times 4$ Labyrinths

As we scale the environment to a $4 \times 4$ labyrinth, the imitation learning task becomes noticeably more challenging. Unlike the $3 \times 3$ case, where the overlap of many trajectory aids generalisation, larger labyrinths induce more structural and strategic diversity, reducing the benefit of purely behavioural memorisation.

Table 4: Results for training, validation and testing splits for the navigation task in a $4 \times 4$ Labyrinth.

| Splits | Metric | BC | DAgger | GAIL | BCO | SQIL | IUPE |
|---|---|---|---|---|---|---|---|
| Train | $AER$ | $-2.29 \pm 2.36$ | $-2.29 \pm 2.36$ | $-2.35 \pm 2.35$ | $-2.29 \pm 2.36$ | $-2.32 \pm 2.32$ | $\mathbf{-1.53 \pm 2.47}$ |
| | $SR$ | $34.00\%$ | $34.00\%$ | $33.00\%$ | $34.00\%$ | $34.00\%$ | $\mathbf{50.00}\%$ |
| Valid. | $AER$ | $-2.29 \pm 2.36$ | $-2.14 \pm 2.4$ | $-2.29 \pm 2.36$ | $-2.29 \pm 2.36$ | $-3.24 \pm 1.77$ | $\mathbf{-0.76 \pm 2.34}$ |
| | $SR$ | $34.00\%$ | $37.00\%$ | $34.00\%$ | $34.00\%$ | $15.00\%$ | $\mathbf{65.00}\%$ |
| Test | $AER$ | $-2.14 \pm 2.4$ | $-2.29 \pm 2.36$ | $-2.20 \pm 2.40$ | $-2.14 \pm 2.4$ | $-3.38 \pm 1.63$ | $\mathbf{-0.61 \pm 2.29}$ |
| | $SR$ | $37.00\%$ | $34.00\%$ | $36.00\%$ | $37.00\%$ | $12.00\%$ | $\mathbf{68.00}\%$ |
| $s_0$ | $AER$ | $-1.82 \pm 2.47$ | $-1.67 \pm 2.48$ | $-1.80 \pm 0.99$ | $\mathbf{-1.51 \pm 2.49}$ | $-2.61 \pm 2.22$ | $-1.53 \pm 2.47$ |
| | $SR$ | $43.00\%$ | $46.00\%$ | $44.00\%$ | $\mathbf{50.00}\%$ | $28.00\%$ | $50.00\%$ |
| $s_0$ and $g$ | $AER$ | $-3.69 \pm 1.2$ | $-3.84 \pm 0.87$ | $-3.80 \pm 0.39$ | $-3.84 \pm 0.87$ | $\mathbf{-2.92 \pm 2.04}$ | $-3.07 \pm 1.93$ |
| | $SR$ | $6.00\%$ | $3.00\%$ | $4.00\%$ | $3.00\%$ | $\mathbf{21.00}\%$ | $18.00\%$ |

Table 4 illustrates this transition, with most methods plateauing at relatively low success rates across all splits. Traditional behavioural approaches – BC, DAgger, and GAIL – achieve comparable performance during training ($SR \approx 34\%$) and show minimal gains on validation and test sets, indicating a limited capacity to generalise beyond memorised demonstrations. In contrast, IUPE substantially outperforms all baselines in validation ($65\%$) and test ($68\%$), with consistent superiority across every

evaluation split. This suggests that the inductive biases introduced by IUPE — likely due to its inverse
dynamics structure and self-supervised training scheme yields a more robust understanding of the environment's transition dynamics, enabling better policy adaptation to unseen instances. Notably, even
in the more difficult $s_0$ and $g$ setting, IUPE achieves a $50\%$ $SR$, outperforming all methods except
BCO. Meanwhile, SQIL continues to struggle, especially in generalisation scenarios: it achieves the
lowest SR on validation ($15\%$) and test ($12\%$), although it obtains the highest SR ($21\%$) in the most
challenging $s_0$ and $g$ setting. This anomalous result may be due to the reward-shaping mechanism in
SQIL biasing action selection towards diverse outcomes, which occasionally aligns with goal-directed
behaviour under novel conditions (especially in the case of smaller labyrinths). Overall, Table 4
highlights the fragility of standard imitation learning methods under modestly increased environment
complexity and reinforces the promise of self-supervised and model-based techniques for achieving
broader generalisation.

## 4.3 Results for $5 \times 5$ Labyrinths

The $5 \times 5$ environment represents the most complex setting in our benchmark, with longer optimal
trajectories, sparser rewards, and greater variation in structure.

Table 5: Results for training, validation and testing splits for the navigation task in a $5 \times 5$ Labyrinth.

| Splits | Metric | BC | DAgger | GAIL | BCO | SQIL | IUPE |
|---|---|---|---|---|---|---|---|
| Train | $AER$ | $-2.11 \pm 2.41$ | $-1.18 \pm 2.45$ | $-0.98 \pm 1.89$ | $-0.53 \pm 2.23$ | $-3.80 \pm 0.96$ | $\mathbf{0.27 \pm 2.39}$ |
|  | $SR$ | 37% | 57% | 61% | 70% | 4% | **75%** |
| Valid. | $AER$ | $-3.70 \pm 1.18$ | $-3.75 \pm 1.08$ | $-3.57 \pm 1.58$ | $-3.90 \pm 0.69$ | $-3.95 \pm 0.49$ | $\mathbf{-2.80 \pm 2.12}$ |
|  | $SR$ | 6% | 5% | 9% | 2% | 1% | **21%** |
| Test | $AER$ | $-3.90 \pm 0.70$ | $-3.80 \pm 0.97$ | $-3.85 \pm 0.85$ | $-3.85 \pm 0.85$ | $-4.00 \pm 0.00$ | $\mathbf{-3.85 \pm 1.00}$ |
|  | $SR$ | 2% | 4% | 3% | 3% | 0% | **5%** |
| $s_0$ | $AER$ | $-2.00 \pm 0.98$ | $-0.90 \pm 0.97$ | $-0.60 \pm 0.92$ | $-0.35 \pm 0.88$ | $-3.55 \pm 0.57$ | $\mathbf{0.00 \pm 0.80}$ |
|  | $SR$ | 40.00% | 62.00% | 68.00% | 73.00% | 9.00% | **80.00%** |
| $s_0$ and $g$ | $AER$ | $-4.00 \pm 0.00$ | $-4.00 \pm 0.00$ | $-4.00 \pm 0.00$ | $-4.00 \pm 0.00$ | $-4.00 \pm 0.00$ | $-4.00 \pm 0.00$ |
|  | $SR$ | 0.00% | 0.00% | 0.00% | 0.00% | 0.00% | 0.00% |

As shown in Table 5, this setting induces significant degradation in performance across all methods,
especially in generalisation. Training success rates remain relatively high for methods such as IUPE
($75\%$) and BCO ($70\%$), indicating that their self-supervised approach is capable of fitting the training
distribution well. However, this performance does not transfer to unseen configurations: validation
success rates drop to $21\%$ and $2\%$ respectively, and test results fall even further, with IUPE reaching
only $5\%$ and BCO stagnating at $3\%$. This steep decline underscores the increasing gap between
memorisation and generalisation as environment complexity grows.

Most striking is the complete failure of all the methods in the $s_0$ and $g$ split, where the success rates
uniformly drop to $0\%$. This clearly illustrates that none of the evaluated methods, including those
with stronger inductive structures such as IUPE or BCO, can act meaningfully when both the initial
and goal states are novel. Although IUPE achieves the highest SR in the $s_0$ setting ($80\%$), indicating
a capacity to generalise from unseen starting positions alone, its inability to deal with novel goals
further confirms the limitations of current methods in reasoning about goal-conditioned policies. The
failure of SQIL is especially significant, performing poorly across all splits, achieving just $4\%$ in
training and $\approx 0\%$ elsewhere, including both generalisation conditions. This suggests that its reliance
on implicit Q-learning from suboptimal reward signals is particularly ill-suited for environments with
high state aliasing and long planning horizons.

## 4.4 Extra Results Conclusion

Across the three levels of environment complexity, a consistent trend emerges. Although most
methods perform reasonably well under training conditions and modest distribution shifts, their
generalisation rapidly degrades as task structure becomes more diverse. In the simplest $3 \times 3$ setting,
even behavioural approaches perform competitively; but by the time we reach the $5 \times 5$ environments,
success is effectively unattainable for all methods on the most challenging splits.

The strongest generalisation comes from IUPE, followed by BCO. Both of which leverage self-
supervised objectives or explicit modelling of transitions. These methods show meaningful robustness
across multiple splits, especially in intermediate conditions like unseen initial states. Still, none of

the evaluated techniques succeed when initial and goal configurations change, highlighting the lack of robustness of current imitation learning strategies when exposed to true task-level variability.

These findings underscore the pressing need for novel approaches in imitation learning that go beyond local behaviour matching. Promising directions include goal-conditioned modelling, compositional planning, or hybrid methods that blend imitation with exploratory or model-based learning components. Ultimately, our benchmark reveals that while simple environments can be mastered through memorisation and interpolation, scalable generalisation demands a fundamentally different approach.

# 5 Extra Results for Most Used Environments

Table 1 in the main work provides a summary of the first-state distribution across $100,000$ initialisations for common imitation learning environments, using Manhattan distance as a proxy for distributional similarity. These results show that environments like MountainCar, CartPole, and Hopper exhibit extremely low variation in initial states, suggesting that agents are often evaluated under conditions very similar to those seen during training. To complement this, we conducted additional experiments to measure distributional shifts in both action and trajectory (state) distribution between training and evaluation. For discrete environments such as Labyrinth, CartPole, and MountainCar, we used the Jensen-Shannon (JS) distance, which is well-suited for comparing categorical distributions like discrete action spaces. For continuous control environments (e.g., Hopper, Walker, and HalfCheetah), we used the Wasserstein (WS) distance for both action and trajectory distributions, as it better captures geometric differences in continuous spaces. We note that we also use the WS distance for the Labyrinth state space to make the comparison between values fairer.

The results from Table 6 work as a proxy for the heatmap images from Figure 2 from our main work. Moreover, we display the average reward to show that the imitation learning agents are not performing poorly, but rather that they are achieving a reward close to their teachers, and that the low difference is not a result of them failing early in an episode. The table below shows that Labyrinth exhibits significantly higher distributional divergence, with an action distribution difference of $0.0165$ and a trajectory difference of $0.0475$. Compared to the other benchmarks, where values are often near zero, these results reinforce our claim that Labyrinth presents a more rigorous generalisation challenge, even in small maze configurations. While this analysis emphasises statistical significance via p-values, it is important to contextualise these values alongside the magnitude of behavioural change. For example, environments such as CartPole and MountainCar show very low p-values despite minimal trajectory differences ($0.0035$ and $0.0037$, respectively), indicating that even minor deviations are statistically consistent but not necessarily impactful. In contrast, Hopper exhibits both a low trajectory difference ($0.0023$) and a non-significant p-value ($0.924$), suggesting stable behaviour. Labyrinth, however, combines a large trajectory difference with a highly significant p-value, providing strong evidence of meaningful behavioural divergence.

Table 6: Action and trajectory difference over $1 \times 10^5$ seeds.

| Environment | Average Reward | Action Distribution Difference | Average Trajectory Difference | Trajectory p-value |
|---|---|---|---|---|
| Labyrinth | 0.9535 | 0.0165 | 0.0475 | 0.000 |
| MountainCar | -101.1170 | 0.0093 | 0.0035 | 0.011 |
| CartPole | 500.0000 | 0.0000 | 0.0037 | 0.000 |
| Hopper | 3530.6376 | 0.0007 | 0.0023 | 0.924 |
| Walker | 4713.9521 | 0.0005 | 0.0016 | 1.000 |
| HalfCheetah | 9526.0863 | 0.0014 | 0.0103 | 0.254 |

# 6 On the Difference from Procedurally Generated Datasets

Machine learning and reinforcement learning benefit from standardised benchmarking datasets and environments, which enable consistent comparisons across methods and ensure reproducibility. These benchmarks often include hidden test sets to preserve evaluation integrity. In contrast, imitation learning lacks such standardisation. Imitation learning studies frequently rely on procedurally generated datasets, which cannot be fully reproduced, making it impossible to compare results fairly. Researchers often need to generate new teacher datasets, locate or reimplement baseline methods, and tune hyperparameters, all of which introduce variability and potential bias. This lack of uniform

benchmarks complicates evaluation and reinforces the existing challenges in imitation learning testing practices.

Environments, such as MiniGrid [2], Procgen [3], and Crafter [5], are valuable for evaluating generalisation in reinforcement learning and, in principle, could be used for imitation learning as well. However, imitation learning faces unique challenges in terms of achieving generalisation, despite using the same environments. This stems from the fact that imitation learning agents are trained solely on expert demonstrations and do not interact with the environment during training (in offline settings). As a result, they lack exposure to out-of-distribution states and must rely entirely on the coverage and diversity of the training trajectories. In contrast, reinforcement learning agents are free to explore and adapt during training, allowing them to recover from unfamiliar states and improve upon suboptimal demonstrations. The difference in training dynamics means that generalisation failures in imitation learning are often more severe and harder to diagnose, especially when the evaluation protocol lacks control over the distributional shift between training and test environments (one of the key differences between Labyrinth and the other environments not specifically designed for imitation learning, such as the ones you mentioned).

Regarding the use of hidden test sets: while reinforcement learning benchmarks like Procgen and Crafter do use held-out seeds for evaluation, they typically sample environments randomly, which can lead to inconsistent generalisation pressure across runs. Labyrinth addresses this by offering explicit control over structural variation, start and goal positions, and task complexity (allowing users to isolate specific generalisation factors and reproduce experiments precisely). Another advantage of Labyrinth over non-imitation learning environments.

To your question about whether there are benchmarks solvable by reinforcement learning but not imitation learning: an illustrative example is the MetaWorld benchmark [7]. MetaWorld comprises a suite of robotic manipulation tasks with procedural variations in object positions and goals. Reinforcement learning agents trained with exploration and reward feedback, such as those using meta-learning or model-based strategies, have demonstrated strong generalisation across unseen tasks (approximately 60% for meta-reinforcement learning according to McLean et al.). In contrast, imitation learning agents trained solely on expert demonstrations often fail to generalise in MetaWorld [8, 9], especially when the demonstrations do not fully cover the diversity of initial states or task configurations. Wan et al. achieved an average of 42% across multiple tasks, with their baselines Behavioural Cloning achieving 34% (half of the reinforcement learning results), and Wei et al. achieved an average around 32%.

Moreover, Wei et al. notes:

> We can infer from this that imitation learning alone may not suffice to build a truly general-purpose model, particularly when aiming to tackle tasks that span a broad range of domains. Even within a single domain, variations in embodiments, scenes, and instructions can pose significant challenges.

We attribute this to the fact that imitation learning agents lack the ability to recover from unfamiliar states or adapt to novel goal during training. For us, MetaWorld exemplifies a benchmark where reinforcement learning succeeds due to its interactive learning paradigm, while imitation learning struggles without sufficient coverage or structural reasoning. This contrast reinforces the motivation behind Labyrinth: to provide a controlled and interpretable environment where generalisation failures in imitation learning can be studied in isolation from exploration and reward shaping.

Finally, Labyrinth and MiniGrid are both grid-based and support procedural generation. We have already pointed out some differences between them, but we can briefly enumerate Labyrinth's unique features as follows:

1. *Full observability and known optimal actions*, enabling fine-grained evaluation and debugging;

2. *Explicit separation of training, validation, and test sets*, with guarantees of structural uniqueness;

3. *Control over generalisation complexity and distribution shifts*, allowing for better evaluation of imitation learning methods generalisation capabilities; and

4. *A reproducible configuration language and dataset generation tools*, tailored for imitation learning benchmarking.

We believe these features make Labyrinth uniquely suited for studying generalisation in imitation learning, while also providing a foundation for reinforcement learning evaluation in future work.