# OpenReview forum: "Quantifying Generalisation in Imitation Learning"
_NeurIPS.cc/2025/Datasets_and_Benchmarks_Track — NeurIPS 2025 Datasets and Benchmarks Track poster_

### Official Review · Reviewer_f9MH · 2025-06-19

**Rating:** 4
**Confidence:** 1

**Summary:**

This paper introduces Labyrinth, a new benchmarking environment for imitation learning aimed at rigorously evaluating agents’ generalisation capabilities. Labyrinth provides a discrete, fully observable state space with explicit start and goal positions, enabling fine-grained analysis of agent behavior. It supports multiple task variants (e.g., key-and-door, partial observability, ice-floor hazards) and allows controlled, reproducible splits between training, validation, and test environments. The authors benchmark six imitation learning methods across these settings and demonstrate that most struggle to generalise beyond training data. The environment is implemented using gymnasium, supports both image and vector observations, and is accompanied by open-source code and datasets hosted on HuggingFace.

**Dataset Code Accessibility:**

Yes

**Ethical Considerations:**

No, there are no or only very minor ethics concerns

**Final Justification:**

I would like to remain my score as 4.

**Limitations Weaknesses:**

1. Limited support for continuous action spaces:
Labyrinth currently only supports discrete actions, which restricts its applicability to many imitation learning methods designed for continuous control tasks (e.g., those used in robotics or MuJoCo environments).

2. Weak generalisation performance in experiments:
The benchmark results reveal that even with extended training or improved architectures, existing imitation learning methods struggle to generalise in Labyrinth. While this highlights the challenge, it also raises questions about whether the environment may be too difficult or misaligned with current method capabilities.

**Strengths Contributions:**

1. Precise control over generalisation conditions:
Labyrinth allows for systematic variation in environment structure, start/goal positions, and task complexity—enabling targeted evaluation of generalisation, which is rarely feasible in existing imitation learning benchmarks.

2. Fully observable, interpretable environment:
The discrete and fully defined state space, along with known optimal actions, provides a unique opportunity for fine-grained analysis, debugging, and comparison of agent behavior—crucial for evaluating learning beyond memorisation.

3. Strong emphasis on reproducibility and usability:
The benchmark is implemented with gymnasium, includes code, datasets, and tools for generating and saving environments, and supports open evaluation on HuggingFace—ensuring easy adoption and rigorous reproducibility.

---

> ### Author Rebuttal · Authors · 2025-07-29
>
> We thank the reviewer for the thoughtful and constructive feedback. We appreciate your recognition of Labyrinth’s potential to support rigorous, interpretable evaluation in imitation learning, and we value your insights on its current limitations and future directions. Below is our response to all points enumerated in the "Limitations Weaknesses" section in the same order.
>
> > 1. Limited support for continuous action spaces: Labyrinth currently only supports discrete actions, which restricts its applicability to many imitation learning methods designed for continuous control tasks (e.g., those used in robotics or MuJoCo environments).
>
> Q1 We agree that using only discrete actions is a primary limitation of our environment, as mentioned in our conclusion. Support for continuous actions within a continuous state space is under development and will be available in a future release, along with further baselines. At that point, Labyrinth will be the first environment to support both discrete and continuous action and states for benchmarking agents.
>
> > 2. Weak generalisation performance in experiments: The benchmark results reveal that even with extended training or improved architectures, existing imitation learning methods struggle to generalise in Labyrinth. While this highlights the challenge, it also raises questions about whether the environment may be too difficult or misaligned with current method capabilities.
>
> Q2 This is an important question. Our main motivation for creating Labyrinth arose from the concern that current benchmarks may no longer be sufficient to fully evaluate the actual understanding of the task dynamics by existing imitation learning methods. We believe that one of the possible reasons for the lack of generalisation is a tendency to simply mimic the teacher, rather than acquisition of the fundamental understanding of the relationships between states, actions and goals. The nature of Labyrinth's tasks is arguably more complex than other environments', but the user is given a high level of control of the complexity of solving the tasks benchmarked. Enabling more challenging problems is vital to achieve more general imitation learning agents. We hope that an easy-to-use and reproducible environment such as Labyrinth, with publicly available datasets, will help steer the field into truly generalisable imitation learning methods.

---

### Official Review · Reviewer_Xot3 · 2025-07-01

**Rating:** 4
**Confidence:** 3

**Summary:**

This paper introduces a discrete maze benchmark designed to evaluate generalization in imitation learning. By controllably modifying the maze structure and increasing task complexity through factors such as keys and doors, ice, and partial observability, the benchmark can precisely generate out-of-distribution (OOD) mazes. This allows for rigorous testing of whether imitation learning methods extend beyond mere memorization to achieve true generalization.

**Dataset Code Accessibility:**

Yes

**Ethical Considerations:**

No, there are no or only very minor ethics concerns

**Final Justification:**

My primary concerns about the limited evaluation in the discrete setting and the shallow analysis have been addressed. I will therefore recomand an acceptance and raise my score to a borderline accept.

**Limitations Weaknesses:**

1. The benchmark is conducted on small, discrete mazes, which are relatively simple compared to existing modern benchmarks that typically use high-dimensional, rendered visual inputs.

2. Currently, the evaluation is limited to a single averaged setting (Table 2). It would be more informative if the authors provided separate analyses for different conditions—such as partial observability, subgoal dependencies (e.g., key-and-door tasks), and slippery surfaces (ice floor)—to more thoroughly assess how each factor affects generalization.

3. The benchmark currently includes only six methods. Incorporating more advanced or diverse baselines could strengthen the experimental results and provide a more comprehensive comparison.

4. The paper suggests that using a ResNet-18 encoder helps reduce spurious correlations, thereby improving generalization. This is an interesting observation. However, it would be valuable for the authors to systematically investigate how different encoder architectures impact generalization performance.

In conclusion, the paper presents a strong and relevant motivation, and the proposed benchmark is a promising starting point. However, the current evaluation is rather coarse, focusing solely on a single averaged metric (Table 2). The paper could be significantly improved through more fine-grained analyses, such as exploring the role of encoder architectures, assessing the influence of task complexity (e.g., partial observability, subgoal dependencies, ice floors), and providing deeper discussions of the learning methods being evaluated.

**Strengths Contributions:**

1. This benchmark can easily generate out-of-distribution (OOD) mazes by modifying the configs, which is customisable and reproducible.

2. The optimal action can be derived to investigate the agent behaviors, since the mazes are discrete.

3. This paper is well written, clearly presenting its motivations, overall structures and details.

4. The results in table 2 demonstrates that most existing imitation learning methods cannot generalize well at OOD states, arguring for more advanced imitation learning methods to enhance generalization.

---

> ### Author Rebuttal · Authors · 2025-07-29
>
> We thank the reviewer for the thoughtful and constructive feedback. We appreciate your recognition of the benchmark’s clarity, reproducibility, and potential to rigorously test generalization in imitation learning, as well as your thoughtful suggestions for expanding the evaluation and deepening the analysis. Below is our response to all points enumerated in the "Limitations Weaknesses" section in the same order.
>
> > 1. The benchmark is conducted on small, discrete mazes, which are relatively simple compared to existing modern benchmarks that typically use high-dimensional, rendered visual inputs.
>
> Q1 We acknowledge that modern benchmarks often use high-dimensional, rendered inputs. However, such environments can obscure the source of generalisation errors due to visual noise, partial observability, and the lack of ground-truth optimality. While the mazes are small and discrete, this is by design so as to enable precise control over generalisation factors, such as structural variation, start/goal shifts, and task complexity, while maintaining full observability and known optimal actions. Such precise control is essential for isolating and diagnosing generalisation failures in imitation learning. Labyrinth complements these benchmarks by offering a transparent and interpretable testbed where generalisation can be studied in a controlled and reproducible manner. Moreover, even though the labyrinth sizes in the appendix and in the main work are small ($3 \times 3$, $4 \times 4$, and $5 \times 5$) we see a rapid deterioration of the benchmark results as the size of the environment grows. This finding suggests that difficulty in generalisation is not merely a result of complex visual input or large environment size, but rather due to a deeper limitation in current imitation learning methods, which often rely on local behaviour matching. There is a failure to reason about task-level variability such as different start and goal positions or altered transition dynamics.
>
> > 2. Currently, the evaluation is limited to a single averaged setting (Table 2). It would be more informative if the authors provided separate analyses for different conditions—such as partial observability, subgoal dependencies (e.g., key-and-door tasks), and slippery surfaces (ice floor)—to more thoroughly assess how each factor affects generalization.
>
> Q2  We agree that analysing performance under different settings, such as partial observability, subgoal dependencies and safety constraints, is essential for understanding the robustness of imitation learning methods. While Table 2 in the main work presents results for the standard navigation task in a single averaged setting (due to space limitations), we would like to clarify that the supplementary material provides extensive additional evaluations for other conditions, including different initial states and different initial and goal states (conditions that significantly affect the transition dynamics and action distributions). These analyses already demonstrate that even small labyrinths (e.g., $3 \times 3$) pose substantial generalisation challenges, with all methods failing to generalise reliably. We agree that further results for the key-and-door, ice floor, and partial observability settings would enrich the evaluation, and they are part of ongoing work. We stress that Labyrinth is designed to support such modular evaluations, and we plan to release extended benchmarks covering these variants in future updates.
>
> > 3. The benchmark currently includes only six methods. Incorporating more advanced or diverse baselines could strengthen the experimental results and provide a more comprehensive comparison.
>
> Q3 At the time of submission, we focused on a representative set of six methods that span different paradigms (e.g., behavioural cloning, adversarial, and observation-based approaches) to demonstrate Labyrinth’s flexility and usefulness as a generalisation benchmark. We are currently in the process of benchmarking additional, more recent non-continuous methods. In a second stage, we plan to enable continuous actions/states in Labyrinth, making it the first grid-like environment to allow for continuous and discrete actions. This will allow us to benchmark advanced continuous methods such as CILO [1], MAHALO [2], and OPOLO [3]. We hope that by allowing uniform representation of discrete and continuos actions will help us understand whether continuous methods have some benefit over fully discrete ones.
>
> > 4. The paper suggests that using a ResNet-18 encoder helps reduce spurious correlations, thereby improving generalization. This is an interesting observation. However, it would be valuable for the authors to systematically investigate how different encoder architectures impact generalization performance.
>
> Q4 We fully agree that a systematic investigation into how different encoders affect imitation learning performance would be a valuable extension of this work. For this submission, our primary focus was on introducing and validating the Labyrinth environment as a benchmark for generalisation in imitation learning. Given the scope and space constraints, we opted to include a targeted experiment using ResNet-$18$ to illustrate that architectural choices can influence generalisation, particularly by reducing spurious correlations. This result was intended to motivate future work rather than serve as a comprehensive study. We acknowledge that a broader exploration, comparing various encoder architectures across different Labyrinth settings, would strengthen the understanding of how representation learning interacts with structural generalisation. We plan to pursue this direction in future work and believe Labyrinth provides an ideal platform for such investigations due to its controlled structure and interpretability.
>
> ---
> [1] Gavenski, Nathan, et al. "Explorative imitation learning: A path signature approach for continuous environments." ECAI 2024. IOS Press, 2024. 1551-1558.
>
> [2] Li, Anqi, Byron Boots, and Ching-An Cheng. "Mahalo: Unifying offline reinforcement learning and imitation learning from observations." International Conference on Machine Learning. PMLR, 2023.
>
> [3] Zhu, Zhuangdi, et al. "Off-policy imitation learning from observations." Advances in neural information processing systems 33 (2020): 12402-12413.

---

> > ### Comment · Reviewer_Xot3 · 2025-08-02
> > **Thanks for the detailed responses!**
> >
> > Thank the authors for the thorough explanations and discussion. My primary concerns about the limited evaluation in the discrete setting and the shallow analysis have been addressed. I will therefore recomand an acceptance and raise my score to a borderline accept.

---

### Official Review · Reviewer_TvXV · 2025-07-02

**Rating:** 5
**Confidence:** 4

**Summary:**

The paper presents a simple environment containing four different tasks (settings) to benchmark imitation learning algorithms. The paper shows that although the tasks are very simple, most IL algorithms fail to generalize to novel combinations of task components, revealing the limited generalization ability of modern IL algorithms. This is in contrast to existing benchmarks for IL, which exhibit small variability between training and test tasks and thus cannot be used to test generalization abilities.

**Additional Feedback:**

See the "Limitations" section. In addition, interesting extensions to this environment could include games with changing colors or games involving multiple agents.

**Dataset Code Accessibility:**

Yes

**Dataset Code Comments:**

The environment code and dataset for training BC algorithms are available on GitHub and Hugging Face.

**Ethical Considerations:**

No, there are no or only very minor ethics concerns

**Final Justification:**

The authors addressed all my concerns and I maintain my recommendation to accept the paper.

**Limitations Weaknesses:**

Limitations:

* The variability of the environment is somewhat limited. For example, the environment layout is always structured as a maze. Although this enables easy computation of the optimal solution, testing on different structures beyond mazes—which may have multiple optimal solutions—could present interesting scenarios.

* I suggest presenting the comparison to existing methods (lines 235-259) in a table alongside the results from the proposed environment.

* I suggest also mentioning procedurally generated datasets for benchmarking generalization in RL, such as MiniGrid [1], Procgen [2], and Crafter [3], and explaining how they differ from the proposed environment.

Questions:

1) It seems that the environment could also be used to test generalization in RL algorithms, and the problems mentioned in lines 235-259 regarding existing environments for testing generalization also apply to RL algorithms. Am I correct?

2) Figure 2 - Would it be possible to create similar graphs for existing IL benchmarks such as CartPole, MountainCar, or Hopper?

[1] Chevalier-Boisvert, Maxime, et al. "Minigrid & miniworld: Modular & customizable reinforcement learning environments for goal-oriented tasks." Advances in Neural Information Processing Systems 36 (2023): 73383-73394

[2] Cobbe, Karl, et al. "Leveraging procedural generation to benchmark reinforcement learning." International conference on machine learning. PMLR, 2020

[3] Hafner, Danijar. "Benchmarking the spectrum of agent capabilities." arXiv preprint arXiv:2109.06780 (2021)

**Strengths Contributions:**

* The paper presents an important generalization problem in modern IL algorithms and provides a simple benchmark to study this problem.

* The paper is well-written and easy to follow.

* The paper discusses and clearly demonstrates the limitations of existing benchmarks for measuring generalization abilities.

* I appreciate the effort to evaluate existing IL algorithms across different benchmark settings.

* The GitHub repository is well-organized and clearly explains the environment.

* The environment contains many useful features, such as image and state (vector) representations that can be easily modified.

---

> ### Author Rebuttal · Authors · 2025-07-29
>
> We thank the reviewer for the thoughtful and constructive feedback. Your recognition of the benchmark’s clarity, relevance, and potential to reveal generalization challenges in imitation learning is deeply appreciated, and your suggestions for future extensions and comparisons are highly valuable for guiding our next steps. Below we answer all your limitations/questions in the same order they were itemized/enumerated in the "Limitations Weaknesses" section.
>
> > * The variability of the environment is somewhat limited. For example, the environment layout is always structured as a maze. Although this enables easy computation of the optimal solution, testing on different structures beyond mazes—which may have multiple optimal solutions—could present interesting scenarios.
>
> L1 For ease of use, Labyrinth creates a maze by default. However, other types of structures, such as MiniGrid's *''Four Rooms''* and *''Memory''* structures, can be loaded via the load function (from Sec. 2.4). Furthermore, Labyrinth's creation function allows the user to customise the intended number of solutions  ([https://github.com/NathanGavenski/Labyrinth/blob/main/src/labyrinth/labyrinth.py#L575](https://github.com/NathanGavenski/Labyrinth/blob/main/src/labyrinth/labyrinth.py#L575)). For example, the number of solutions of the ice floor structures can be adjusted to suit the user's specific needs (although Labyrinth ensures there are always at least two solutions). However, we do not allow users to specify how many *optimal* solutions each structure should have, but this is a great idea and will add the functionality in future releases.
>
> > * I suggest presenting the comparison to existing methods (lines 235-259) in a table alongside the results from the proposed environment.
>
> L2 We are not sure we understood your suggestion. The results mentioned in 235--259 were obtained by brute-force and do not apply to existing methods. They were merely mentioned to exemplify how generalisation can emerge in some environments without a deeper understanding of their dynamics. We will be happy to clarify further if the suggestion meant something else.
>
> > * I suggest also mentioning procedurally generated datasets for benchmarking generalization in RL, such as MiniGrid [1], Procgen [2], and Crafter [3], and explaining how they differ from the proposed environment.
>
> L3 Great idea, we will discuss the role of procedurally generated datasets and emphasise the distinction between them and what's offered by Labyrinth in the supplementary material. To be clear, machine learning and reinforcement learning benefit from standardised benchmarking datasets and environments, which enable consistent comparisons across methods and ensure reproducibility. These benchmarks often include hidden test sets to preserve evaluation integrity. In contrast, imitation learning lacks such standardisation. Imitation learning studies frequently rely on procedurally generated datasets which cannot be fully reproduced, making it impossible to compare results fairly. Researchers often need to generate new teacher datasets, locate or reimplement baseline methods, and tune hyperparameters, all of which introduce variability and potential bias. This lack of uniform benchmarks complicates evaluation and reinforces the challenges already present in imitation learning testing practices.
>
> > 1. It seems that the environment could also be used to test generalization in RL algorithms, and the problems mentioned in lines 235-259 regarding existing environments for testing generalization also apply to RL algorithms. Am I correct?
>
> Q1. Yes, however, imitation learning in particular suffers from this generalisation issue, as it resides at the intersection of machine learning and reinforcement learning. In traditional machine learning, evaluation is done using a separate test set drawn from the same distribution as the training data, ensuring a clear and standardised assessment. In contrast, imitation learning typically evaluates performance through environment simulations rather than a held-out test set. This is because imitation learning is often applied in agent-based settings where the learned policy is tested by simulating its behaviour. However, unlike reinforcement learning, imitation learning agents are trained only on demonstrated trajectories and lack exposure to states outside those demonstrations. This can lead to poor generalisation, especially if the agent encounters unfamiliar states during testing. Therefore, understanding how the training data relates to the test scenarios is crucial, as mismatches can lead to misleading evaluations -- either by being overly pessimistic due to unseen states or overly optimistic due to data leakage.
>
> > 2. Figure 2 - Would it be possible to create similar graphs for existing IL benchmarks such as CartPole, MountainCar, or Hopper?
>
> Q2. Yes, and indeed we already have them. Unfortunately, due to space constraints we could not include them in the main paper. However, since there is interest, we will be happy to add them to the supplementary material.

---

### Official Review · Reviewer_LsRf · 2025-07-03

**Rating:** 5
**Confidence:** 4

**Summary:**

This paper introduce Labyrinth, a benchmarking environment designed to test generalisation with precise control over structure. Labyrinth extends gridworld with more configurable structures and computable optimal actions for given states. The authors collected a set of training data for imitation learning and verify that the environment is indeed challenging and useful for imitation learning.

**Dataset Code Accessibility:**

Yes

**Ethical Considerations:**

No, there are no or only very minor ethics concerns

**Final Justification:**

The authors addressed my concerns and I will keep my current score.

**Limitations Weaknesses:**

1. line 67 "agents cannot perform67 state-matching by forcing a path to be similar to its training data" Why is this true? If the task is similar, the agent can still follow the same strategy in the training data, no? Is this because the training env and testing env are different? But shouldn't training configurations that are similar with the test configurations give some information as well?
2. the contribution of the proposed training environment seems to be limited, due to the existence of the [MiniGrid environment](https://minigrid.farama.org), which includes POMDPs, door-key, walls, even with lava (trap states) and multi-room settings. I do recognize the effort of the authors to make the environment much more comprehensive than MiniGrid. However, the varity of the training environments are less than that of MiniGrid, which includes more environmental elements and can be more realistic (such as the multi-room setting).
3. The highlighted contribution of this work is that we can compute the optimal action under any configurations. I think that is quite interesting, but I did not see enough discussion on that.

**Strengths Contributions:**

Overall I find the proposed environment is very interesting and can be very helpful. Often times practioners find themselves cannot explain why the imitation learning agent they trained perform in a particular way, or verify that whether it is indeed optimal. This work can help shed light on some of the issues.
1. The proposed training environment is comprehensive and challenging, yet configurable and interpretable. This can be a good tool for analyzing the algorithm performance for imitation learning.
2. I find the structure of the environment simple and intuitive. At each configuration we are able to compute the optimal action, which is useful for interpretation.
3. The setting for partial observability is practical and useful for testing agents that will be deployed in real life, which are often POMDPs.

---

> ### Author Rebuttal · Authors · 2025-07-29
>
> We thank the reviewer for the thoughtful and constructive feedback. The recognition of Labyrinth’s potential to enhance interpretability and provide a rigorous, configurable benchmark for imitation learning is especially encouraging. Below we answer all your questions in the same order they were enumerated in the "Limitations Weaknesses" section.
>
> > 1. line 67 "agents cannot perform67 state-matching by forcing a path to be similar to its training data" Why is this true? If the task is similar, the agent can still follow the same strategy in the training data, no? Is this because the training env and testing env are different? But shouldn't training configurations that are similar with the test configurations give some information as well?
>
> Q1. You are correct, imitation learning systems should leverage training information to behave similarly to the teacher and perform better during testing. However, during our experiments, we discovered that when solving a labyrinth, all agents would predict the action according to the proximity of their current position to a position seen in the training data and the similarity of the structures near them. In other words, when retrieving the image in the training data closest to the current state during testing, the agent sometimes would not match their position within the maze in favour of a position with the most similar structure. Yet, sometimes the agent would retrieve a training image that would not match the current structure (e.g., there would be differences between the walls in the training and current position) and would predict an action that would move them towards a wall (yielding no transition) causing them to get stuck. In other environments, such as MuJoCo, imitation learning agents can force a path by taking the agent to a suboptimal state during early stages and then using state-matching to achieve optimal results. This behaviour is impossible in a structure like Labyrinth, where walls in the current state may prevent agents from reaching a state present in the training data. Navigating the Labyrinth requires a deeper understanding of the correlation between the agent's current position, wall structure, and optimal action choice. We reached the above understanding after experiments conducted with the use of Labyrinth, that unfortunately did not make it in time to be included in this submission.
>
> > the contribution of the proposed training environment seems to be limited, due to the existence of the MiniGrid environment, which includes POMDPs, door-key, walls, even with lava (trap states) and multi-room settings. I do recognize the effort of the authors to make the environment much more comprehensive than MiniGrid. However, the varity of the training environments are less than that of MiniGrid, which includes more environmental elements and can be more realistic (such as the multi-room setting).
>
> Q2. We agree that, for now, Labyrinth provides less variability than other environments, such as MiniGrid. However, our focus with this environment was on the reproducibility and interpretability of the results of benchmarking, as well as easy customisation of the experiments for users. Although Labyrinth can be used for evaluation of methods in other fields, it was specifically designed and optimised for imitation learning, which means that it offers datasets with clear splits between training, evaluation and test sets. This allows methods to be benchmarked according to the exact same settings, ensuring fairer comparison and fully reproducible results. Labyrinth also offers better interpretability by allowing users to debug agents and compare their action choices with the optimal action on every path, as the environment provides all possible solutions given a predefined structure. Labyrinth offers an easy and fully customisable environment for all tasks via a definition language and allows users to save and later load the created tasks. If the user desires to understand how minor changes in a maze's structure or task impact their agent, they can easily save it and modify it manually or programmatically to compare the performance --- a feature that MiniGrid lacks. Finally, we are currently working on implementing a custom type of tile with user-defined behaviour to be used in new tasks. This change will allow a significant increase in the number of possible tasks supported by the environment beyond the ones already defined by default, allowing for all MiniGrid tasks to be implemented. In hindsight, we agree that a comparison with other environments would have stressed the critical differences between them, making it easier for users to pick the best option for their evaluations, and we will include the comparison in the final version of our submission.
>
> > The highlighted contribution of this work is that we can compute the optimal action under any configurations. I think that is quite interesting, but I did not see enough discussion on that.
>
> Q3. We hope that the response provided to Q2 above allays the reviewer's concerns about our claim, and in light of the clarifications provided, we will improve the narrative in the paper around this.

---

### Decision · Program_Chairs · 2025-09-18

**Decision:**

Accept (poster)

**Comment:**

This paper proposes a new benchmarking environment designed to rigorously test generalisation in imitation learning. The contributions consist of a precisely controllable environment, generalisation metrics for quantifying performance in this environment, and an empirical study of different imitation learning methods on this benchmark. Reviewers find the benchmark useful; they highlight it's fine-grained analysis, reproducibility, and clarity. The most important concerns surfaced during the reviewing process were around the limited variety in this benchmark compared to alternatives, and the lack of connections to real-world imitation learning tasks.The authors clarified key points of discussion, and they provided additional experimental results. The reviewers are largely satisfied with the authors' response, with ratings ranging from borderline accept to accept.
I recommend accepting this paper, and suggest the authors further incorporate the reviewers' suggestions to increase the paper's scope and deepen its analysis.